# Mold in Paradise: A Review of Fungi Found in Libraries

**DOI:** 10.3390/jof9111061

**Published:** 2023-10-30

**Authors:** Islam El Jaddaoui, Hassan Ghazal, Joan W. Bennett

**Affiliations:** 1Laboratory of Human Pathologies Biology, Department of Biology, Faculty of Sciences, University Mohammed V, Rabat 10000, Morocco; 2Genomic Center of Human Pathologies, Faculty of Medicine and Pharmacy, University Mohammed V, Rabat 10000, Morocco; 3Department of Plant Biology, Rutgers, The State University of New Jersey, New Brunswick, NJ 08901, USA; profmycogirl@yahoo.com; 4Laboratory of Genomics and Bioinformatics, School of Pharmacy, Mohammed VI University of Health Sciences, Casablanca 82403, Morocco; hassan.ghazal@fulbrightmail.org; 5Royal Institute of Sports, Royal Institute for Managerial Training in Youth and Sport, Department of Sports Sciences, Laboratory of Sports Sciences and Performance Optimization, Salé 10102, Morocco

**Keywords:** library, filamentous fungi, mold, mould, mold isolation, biodeterioration, indoor microbiomes, *Aspergillus*

## Abstract

Libraries contain a large amount of organic material, frequently stored with inadequate climate control; thus, mold growth represents a considerable threat to library buildings and their contents. In this essay, we review published papers that have isolated microscopic fungi from library books, shelving, walls, and other surfaces, as well as from air samples within library buildings. Our literature search found 54 published studies about mold in libraries, 53 of which identified fungi to genus and/or species. In 28 of the 53 studies, *Aspergillus* was the single most common genus isolated from libraries. Most of these studies used traditional culture and microscopic methods for identifying the fungi. Mold damage to books and archival holdings causes biodeterioration of valuable educational and cultural resources. Exposure to molds may also be correlated with negative health effects in both patrons and librarians, so there are legitimate concerns about the dangers of contact with high levels of fungal contamination. Microbiologists are frequently called upon to help librarians after flooding and other events that bring water into library settings. This review can help guide microbiologists to choose appropriate protocols for the isolation and identification of mold in libraries and be a resource for librarians who are not usually trained in building science to manage the threat molds can pose to library holdings.

## 1. Introduction

When we speak of libraries, we mean both the books, documents, and other materials that constitute a collection as well as the buildings that house these materials. Perhaps the most famous quote about libraries comes from Argentinian author Jorge Luis Borges (1899–1986), who said, “I have always imagined that Paradise will be a kind of library.” (“Siempre imaginé que el Paraíso sería algún tipo de biblioteca.”) [1].

Throughout history, the biggest threat to libraries has come from fires [2]. These fires have been a side effect of war, set deliberately by conquerors, or as collateral damage during bombing. Sometimes, religious fanatics who have feared the secular knowledge stored within libraries create purposeful conflagrations; in other cases, fires occur through accidents or poor management of the physical plants in which library collections are housed [3,4,5].

This review focuses on another threat to libraries that have received far less attention than fire, namely, the insidious growth of molds, mildews, and other forms of microbial life. Fungal damage to library materials is often first suspected because of the musty odors that are caused by volatile organic compounds (VOCs) produced during the biodeterioration process (Figure 1) [6]. Fungi also increase the rate of paper deterioration through foxing. The color of the fox’s spots can originate from two sources: a rust-red alkali-soluble material and an alkali-insoluble straw-colored stain in the paper fibers. When contaminated materials are new, there is no noticeable discoloration, but the fungal structures or products form a suitable chemical environment that, over time, causes discoloration [7]. More dramatically, lush and easily observed mold growth can be a serious aftereffect of fire events. When fires are discovered early enough for fire departments to succeed in putting them out, in some cases, the ensuing water and microbial damage constitutes a major part of the loss of books [8]. Most commonly, however, water in libraries is due to intrusion from floods, structural leaks, or condensation when moist warm air encounters a cold surface. In humid climates where air conditioning is widespread, condensation can be a major problem despite good building maintenance. Outdoor fungal aerosols often dominate indoor fungal populations, even with complex ventilation systems. When sufficient water is present, dust that accumulates in carpeting and ventilation duct surfaces can become important inoculum sources [9,10].

Climate change and global warming have increased the threat of extreme storms, flooding, and other weather events that lead to water incursion in buildings. Although libraries constitute only one type of built environment, their contents make them particularly susceptible to microbial biodeterioration. In addition to paper, volume bindings made of cardboard, fabric, leather, and glue are all good substrates for mold growth [11,12].

Not only do fungi represent a threat to books in libraries, but they may also be a threat to librarians and patrons. Mold growth in indoor environments has been consistently associated with negative health outcomes, including allergies, respiratory symptoms (especially rhinitis), exacerbation of asthma, and even hypersensitivity pneumonitis [13,14,15]. More controversially, many people who have been exposed to high levels of mold experience various non-specific symptoms, including extreme fatigue, headaches, debilitating respiratory problems, and so forth, in an illness often called “sick building syndrome” [16]. This controversial diagnosis has received attention from clinicians and members of the legal, worker’s compensation, and building engineering sectors [17,18,19]. Since health-related research attracts most of the funding, research on indoor molds has focused on their potential to harm human health. It is worth noting that librarians and archivists are at higher risk of occupational high exposure levels than almost any other office workers. Nevertheless, in this paper, we focus on the kinds of fungi that have been associated with the biodeterioration of library contents and refer readers to the voluminous published literature on indoor air quality and human health [13,14,20].

## 2. Methods for Sampling and Identification of Fungi

### 2.1. Sampling Methods

The methods used for sampling indoor fungi have been developed and most widely used in the food and pharmaceutical industries or hospital settings [21] and then adapted for use in other indoor environments. The manual published by the American Industrial Hygiene Association (AIHA) titled “Recognition, Evaluation and Control of Indoor Mold” [22] is one of the best single sources for guidance on the evaluation and analysis of indoor molds. Other valuable resources include Chin and Heinsohn [23], Flannigan et al. [24], Verdier et al. [25], and Martin-Sanchez et al. [26].

Many different methods developed for the study of indoor environments have been used to assess mold damage in libraries (Figure 2). These include culturing, mechanical air samplers, spore counting, direct microscopy, and DNA and chemical analyses [22,24,25,26]. Traditional sampling is conducted either from the air, dust, books, shelving, and other surfaces or—when evident—directly from fungal growth [27]. A diagrammatic overview of different traditional sampling techniques is given in Figure 3.

#### 2.1.1. Air Sampling

Choosing the proper air sampling strategy depends on several factors, including indoor environmental conditions, the purpose of the research and analysis to be performed, and the concentration of microorganisms in the air. Most active air sampling equipment consists of a minimum of a sampling pump with an airflow inlet and includes the use of a collection medium. Air sampling also relies on certain parameters such as constant and appropriate flow rates and stable ambient conditions [28]. With culture-based air sampling, the concentrated particulates are then impacted onto an agar growth medium. This sampling method is widely used for determining the presence and concentration of culturable fungal spores and fragments. Quantifiable data from viable air sampling methods are reported as colony-forming units (CFUs) per unit volume of air sampled.

Specialized devices have been developed for the collection of fungal propagules from the air. In a standard impactor sampler, the air is accelerated through a perforated plate or a narrow slit after being drawn into a sampling head with a pump or fan. This creates laminar airflow over the collecting surface, which is frequently a standard agar plate or contact plate filled with suitable agar medium. The size of the pores in sieve samplers and the width of the slit in slit samplers affect the air’s velocity. When air hits the surface of the collection, it produces a tangential variation in direction, and any suspended particles are carried away by inertia, impacting the surface of the collection [29]. A known volume of air is passed through the device at a defined flow rate for a predetermined amount of time. The prototype for this form of sampling was developed by Andersen in 1958 (Ariel A. Andersen, Provo, UT, USA), and many subsequent refinements and improvements have been made since then [22]. A wide range of devices have been developed using the impaction principle. The Andersen sampler, a multi-stage cascade sieve device that uses perforated plates with progressively smaller holes at each level, is one of the most commonly used. This instrument allows particles to be separated based on their size [28].

Impingement-based methods work almost the same as impact-based techniques, except that the microorganisms are collected in a liquid medium. Typically, air is drawn through a narrow inlet tube over the collection medium, where the flow rate of sampled air depends on the diameter of the inlet nozzle. The suspended particles are projected onto the collecting liquid as soon as the air hits the liquid. At the end of sampling, aliquots are grown on appropriate growth media to enumerate viable microorganisms [30]. Impinger sampling is particularly useful for sampling heavily contaminated air because liquid samples can be diluted to the appropriate level for subsequent growth culture analysis [31]. Liquid impingers (The United States Bureau of Mines Experiment Station, Pittsburgh, PA, USA) are more practical for post-analysis, as biological agents present in liquid media would not be dehydrated. However, the minimum particle cutoff size is 300 nm for commercial impingers, and they have shown very low collection efficiency for particles smaller than 100 nm [32].

Filtration-based samplers are relatively less complicated and less expensive for sampling bioaerosols. In the filtration method, airborne microorganisms are collected by passing air through porous membrane filters made of fiberglass, polycarbonate, polyvinyl chloride, or cellulose acetate and incubated by transfer onto the surfaces of growth agar medium or in gelatin [31].

Spore traps are used for non-culture-based analysis. Like culture-based fungal air sampling, these devices rely on the impaction of suspended airborne particles moving at a relatively high velocity. Rather than a growth medium, glass slides with a sticky transparent material or a sticky transparent membrane are used to collect the spores, which can then be counted and examined under an optical microscope [22]. The use of spore-trapping samplers to collect fungal spores, followed by direct microscopy to identify and enumerate the spores, facilitates an estimate of the total number of particulates present in a sample. Sometimes, microscopic stains such as basic fuchsin or lactophenol cotton blue are used to discriminate spores from simple debris [33]. The Burkard spore trap (Rothamsted Experimental Station, Harpenden, UK) is widely used, and samples collected with this instrument are analyzed using microscopy for spore identification, usually to the genus level [34]. However, because only a small number of fungal spore types have distinctive morphologies, this approach only gives a rough idea of the kinds of fungi that are present. *Aspergillus* and *Penicillium* are often the predominant fungi detected from indoor air, including that of libraries. Under the light of a microscope, the spores of both genera are seen as small undifferentiated spheres. Thus, non-culture-based microscopic methods do not allow for the differentiation between genera of *Aspergillus* and *Penicillium* or within species of either genus. In other cases, fungi can be identified, at least to genera, by their spore shape. *Cladosporium*, *Alternaria*, and *Stachybotrys* are examples of genera with distinctive spore morphology. For taxa with morphologically uninformative spore morphology, culture-based methods offer a practical and convenient way to observe enough characters to identify species [35].

#### 2.1.2. Surface Sampling

For surface sampling, swab sampling is the most commonly used. In this method, a sterile swab is rubbed over the surface of interest using a twisting motion. The swab is then subjected to microbial examination, usually by shaking it in a diluent and examining the resulting microbial suspension using pour plate cultures. Swabs are convenient for use on irregular surfaces, where contact plates cannot be used [36].

A contact plate or Rodac (VWR International, Radnor, PA, USA) is an agar poured into a contact plate usually used for surface sample testing. The process begins by removing the lid and gently rolling the agar surface across the sample area, transferring any microorganisms present on the surface to the agar. After obtaining the sample, the cover is replaced, and the surface is cleaned with a wipe and isopropyl alcohol to remove any agar residue remaining from the contact plate. The plates are then incubated at temperatures suitable for the growth of a variety of microorganisms [36].

Adhesive tape (IECL, Brisbane, Australia) sampling is a non-destructive technique that allows the determination of which fungal particles have accumulated on a specific surface. Although tape sampling and direct microscopic examination cannot determine which spores are viable, they can identify some species regardless of their capacity to develop on a specific culture media. To aid proper observation of the sample, the piece of tape must be placed in the center of the microscope slide [22]. This is a simple and inexpensive technique that does not require any specific equipment and shows the existing relationships between the surface and the colonizing microorganisms, the diffusion, and the correspondence with a particular alteration of the surface.

#### 2.1.3. Factors Affecting Sampling

The season of the year, building characteristics such as construction materials, air conditioning system, ventilation, and light source as well as other environmental parameters, mainly indoor temperature and relative humidity, can affect the data collected [37]. The main fungi recovered from a particular environment will partially reflect the analytical method used to detect them [38]. Each sampling strategy and analytical method inevitably involves creating a bias. Fungal counts in floor dust are less affected by short-term variability than airborne concentrations and thus have the advantage of providing an estimate of mold quantities over a given period. Furthermore, certain mechanical samplers are better for certain purposes. The Andersen six-stage sampler is preferred for viable counts, and the Burkard 24 h sampler is preferred for total counts [39]. When studying viable counts, different types of media recover different types of organisms. For example, when media with low water activity are used, they will select for xerophilic fungi [40]. Moreover, all sampling methods based on cultivation will find only viable fungal spores. In the 21st century, PCR has emerged as an effective way of detecting slow-growing species as well as non-viable propagules. A gradual shift toward DNA-based methods has taken place in recent decades [26,41].

#### 2.1.4. Culture-Based and Non-Culture-Based Sampling

The major disadvantage of culture-based sampling over non-culture-based sampling is that it tests only for viable propagules; moreover, not all viable fungal propagules necessarily germinate on the selected growth medium. Certain fungi grow quickly, while others grow slowly. It is possible that one species of quickly growing fungus will dominate and overgrow a nutrient agar plate before other slow-growing types can develop. This means that interpreting data from such samples can be difficult. More importantly, culturable organisms comprise a small fraction of the total number of fungi found in each sample [35,42]. Despite these limitations, culture-based fungal air sampling is still a widespread and useful investigative tool [22]. Moreover, by combining techniques, some of the limits of each methodology are addressed, and many researchers use at least two approaches for any given sample [43]. The paper by Nazaroff [44] provides useful insights into the dynamic processes that govern bioaerosol behavior and possible impacts on human health.

### 2.2. Fungal Identification Techniques

#### 2.2.1. Microscopic Identification

There is a shortage of skilled mycologists who can accurately identify species using traditional microscopic and cultural approaches. Environmental microbiologists have developed alternatives to counting and culturing for the identification of fungi. These include the detection of ergosterol content [45], beta-glucans [46], nucleic acids [47], and VOCs [48] as well as immunoassays [49]. In the majority of cases, molds are identified based on colony and microscopic characteristics. In-depth microscopic analysis of a fungal specimen yields a clear picture and useful details on the fungal structure. Thus, experts can typically identify molds with certainty based on their microscopic structures and morphological characteristics, such as type, size, shape, and arrangement of their spores, as well as their hyphae’s size, color, and septation. However, unlike bacterial and yeast identification, species-level mold identification has not been subject to the same formal standards. A combination of macroscopic and microscopic examination can be used to identify molds. The first step in the identification of fungal specimens is the use of a definite fungal stain. The selection of the staining method is primarily based on the sample under study [50].

#### 2.2.2. Molecular Identification

The single best substitute for counting viable and non-viable colonies is PCR because this technique offers several advantages over the more time-consuming traditional monitoring strategies. It can usually identify fungi to the species level while also decreasing the time required for the more time-consuming and less sensitive culture and counting methods. In addition, quantitative PCR can reduce the variability associated with culture-based methods [51]. Mycologists have developed a system of “DNA barcoding” that facilitates the rapid identification of fungi [52]. Using such culture-independent nucleic acid analysis, Amend et al. [53] found a high diversity of fungi from global indoor environments with *Alternaria*, *Cladosporium*, and *Epicoccum* as well-represented as the *Aspergillus* and *Penicillium* species that tend to dominate culture-based surveys. When several different indoor buildings in Helsinki were studied with this approach, 585 fungi were identified, a much larger number than is usually reported using traditional cultural and microscopic analysis indicating that there is more diversity in indoor fungal populations than has been previously identified [54].

Two primary methods can be used in real-time PCR and traditional PCR to identify the precise target sequences required for the detection and identification of fungal strains: (1) using universal primers to amplify and sequence conserved genes shared by all fungus and (2) using non-specific random primers to amplify unknown genomic regions. The highly stable, varied, and conserved sequences of the nuclear-encoded ribosomal RNA genes (rRNA), as well as their ability to be amplified and sequenced using universal primers, make them desirable targets. They also occur in multiple copies arranged in tandem repeats, with each repeat made up of the 18S small subunit (SSU), 5.8S large subunit, and 28S large subunit (LSU) genes separated by internal transcribed spacer (ITS) regions (ITS1 and ITS2). Another part of rDNA is the spacer between the LSU and SSU genes, called the nontranscribed spacer (NTS) or intergenic spacer (IGS). The rDNA-conserved regions enable the detection of similar ribosomal genes from various species, genera, families, or even kingdoms using probes or primers from a single species. However, there are numerous sequence variations among the identical DNA fragments found in different organisms that might be used to identify them [55]. The key fungal DNA barcode for species identification is the ITS region. The primary benefit of the ITS region is the simplicity of PCR amplification from small samples using universal primers. Furthermore, the ITS region has generally high PCR amplification efficacy rates. Another significant advantage of the ITS is the abundance of high-quality reference sequences that have been deposited in numerous online databases [56]. The majority of sequence variation in rDNA exists within the IGS regions. IGS regions are more challenging to amplify and sequence than ITS regions, but they can be helpful when there are not enough variations among ITS regions. Ascomycete mating-type genes, the elictin, and the laccase gene are other genes that are used as detection targets. Specific target sequences can also be identified by amplifying random regions of the fungal genome with PCR-based strategies such as arbitrarily primed PCR (AP-PCR) and randomly amplified polymorphic DNA (RAPD). Amplified PCR fragments are separated with gel electrophoresis to identify unique PCR bands that can be purified, cloned, and sequenced. Compared with the amplification of conserved genes, this method requires more time and expertise because it necessitates the study of several isolates from both the target species and its related species [55].

While we recognize that until recently, these methods were rarely used in the context of libraries [57], we recommend that in the future, librarians who seek to learn about mold contaminants in their collections should use molecular methods. In the future, DNA-based approaches will become the “gold standard” and will allow more precise and accurate identification of fungal taxa. High-throughput DNA sequencing can facilitate a better characterization of microbial communities. Sequence-based approaches can be used to extend or replace culture- or microscopy-based techniques for identifying indoor fungi and can often give direct identification of species [58]. Ideally, a combination of molecular and morphological techniques to provide accurate species identifications will enable researchers to better evaluate the potential for toxin and allergen production [59]. Obviously, there is a great need for standardizing the methods used for assessing microbial contamination in the built environment. Because so many different approaches are used by different laboratories in different countries, it is difficult to compare findings from different laboratories.

#### 2.2.3. Fourier Transform Infrared (FTIR) Spectroscopy Identification

Conventional phenotypic methods for fungal identification are time-consuming, and most alternative methods are still expensive and call for specialized laboratory knowledge. It has been demonstrated during the past two decades that FTIR spectroscopy is a useful approach for identifying microorganisms. FTIR is a non-destructive, robust [60], and vibrational spectroscopic technique based on measuring the fundamental vibrational modes of a molecular bond. In this method, a sample interacts with a polychromatic infrared source, and the molecules therein can either absorb or reflect the light, stimulating vibrational motions [61]. The foundation of FTIR is the capture of each microorganism’s unique spectral signature after cultivation under predetermined conditions. This spectral signature primarily reflects the biomass’s composition in terms of proteins, lipids, nucleic acids, and carbohydrates [62]. The research conducted by Lecellier et al. [61] validates the efficacy of high throughput FTIR spectroscopy for the identification of fungi utilizing a library of molds with industrial significance [61]. Foxing spots were also identified using FTIR spectroscopy on paper from nine printed books that ranged in age from the early 19th to the mid-20th century [60].

#### 2.2.4. VOCs Identification

As we already mentioned, mycologists also have used VOCs to detect molds. VOCs are carbon-containing compounds of low molecular mass that easily evaporate at normal temperature and pressure. Most microbial VOCs have distinct smells. More than 300 VOCs from microscopic and macroscopic fungi have been reported, and it is usual for dozens, if not hundreds, of fungal species to release the same chemical type of VOC [63]. VOCs are metabolic byproducts of fungi that can be detected before apparent symptoms of microbial growth [64]. As a result, this fingerprint can act as early warning signs of potential biocontamination issues in library materials. The analysis of VOCs produced by molds developed in libraries has been addressed by several studies [65,66,67].

#### 2.2.5. Scanning Electron Microscopy (SEM) Identification 

In addition to VOCs, SEM and energy-dispersive X-ray spectroscopy (EDS) are also used as fungi detection techniques. SEM/EDS comprise what has long been considered the advanced surface examination tool for materials scientists. SEM is the imaging portion of the technique. A scanning electron microscope, in contrast with a standard optical microscope, converts electron interactions into an optical signal using electrons. Despite the benefits of optical microscopy for some applications, there may be resolution restrictions caused by limited focus depth and light wavelength [68]. For the investigation of fungi, the scanning electron microscope is perfectly suited for the observation of intact spore structures over a wide range of magnifications, duplicating and supplementing data obtained using light microscopy [69].

#### 2.2.6. Biochemical Identification

In certain cases, fungal identification may require biochemical tests that distinguish genera among families and species among genera. For the identification of various molds, numerous common biochemical assays are available, such as urease production and proteolysis. When fungi grow in selective solid or liquid media, they ferment carbohydrates and produce alcohols, acids, gases, and enzymatic and metabolic products in patterns characteristic of their genus and/or species. These fermentation byproducts can differentiate between taxa [50].

## 3. Mold Prevention in Libraries

The protection of library materials can be optimized by promoting dry, stable environmental conditions, which, in most climates, means temperature-controlled heating and air conditioning systems. Suitable environmental parameters include setting the temperature as low as possible to reduce temperature-induced deterioration and controlling relative humidity to avert damage to the collection [70,71]. Ideally, each library should have a temperature and humidity monitoring system because if one of these parameters in the building falls outside acceptable limits, mold will grow on library materials [72]. Norms for indoor relative humidity and temperature broadly depend on local climatic conditions, which is why these norms cannot always be considered standards. It is crucial that librarians are aware of the different prerequisites. It is also important to know that a well-installed and maintained air conditioning system is not enough, because the system itself cannot compensate for design and construction defects. The structure of a building must create an airtight environment that is then complemented with an air conditioning system. Implementation of a Heating, ventilation, and air conditioning (HVAC) system sometimes has negative effects, especially with relative humidity [73]. Happily, monitoring library temperature and humidity can be accomplished with a very reasonable investment in a digital thermohygrometer [72].

For libraries in many tropical countries, these climatic control systems are too expensive, leading to a periodic appearance of mold growth [74]. Moreover, in subtropical and tropical climates, air conditioning is usually installed more for human comfort than for the protection of library collections. When these libraries are closed, the systems are often turned off, resulting in wide temperature fluctuations that can lead to water condensation and subsequent mold growth. Several experts have emphasized that maintaining the indoor environment at certain temperatures and relative humidity levels is effective in reducing the propagation of fungi and other microorganisms [37,75,76]. Ventilation lowers levels of existing mold spores in the air while keeping the environment dry and cool. HVAC systems with good system design can provide environmental control over large regions of a building. Proper equipment maintenance also decreases the likelihood of issues caused by system failures [77].

Control of indoor mold is a multidisciplinary problem. Many environmental variables, including occupancy, building structure, maintenance, ventilation, and climate, affect the mycology of a given library. Mycologists should become more aware of the pitfalls associated with the traditional methods of screening for fungi and the challenges that accompany attempts to compare studies across climates, library characteristics, sampling methods, and the like. Proper use and monitoring of HVAC systems minimize mold contamination problems, but the single most important preventative measure is keeping water out by ensuring that roofs do not leak and that indoor plumbing remains in good repair. Librarians should become more aware of the guidelines issued by the “American Society of Heating, Refrigerating and Air-conditioning Engineers (ASHRAE)” and the “Commission of European Communities (CEC) ventilation guidelines”. Over the last half-century, there has been an increasing understanding of the relative contributions of particulate matter, organic fumes, and microbial contamination to human health issues inside residences, schools, commercial buildings, and other indoor environments [20].

## 4. Literature Survey of Molds Found in Libraries

The publications we reviewed were collected with a keyword search in PubMed and Google Scholar using terms such as ‘mold/mould in libraries’; ‘fungal contamination of libraries’; ‘mycology of libraries’; and ‘biodeterioration of books in library environment’. Once papers were identified using the keyword search, additional papers were selected from the references cited in the identified articles. All published studies that examined fungi colonizing library environments were included without making exclusions based on the sample(s) studied, method of analysis, or geographical location. These published studies were divided into three broad categories based on sampling strategies. Table 1 encompasses studies where sampling was conducted only from surfaces and dust. In Table 2, the sampling was performed on air samples only. Table 3 includes studies with mixed sampling strategies from surfaces and/or dust and air. Only the three main genera of fungi identified in each study are listed in the tables. A study by Kalwasińska et al. [78] was not included in these tables because fungi were merely counted to CFUs but not identified to genus and species. Important findings related to specific papers are highlighted in the results section below.

## 5. Results of the Literature Survey

We identified 53 studies in which fungi had been identified to the genus and/or species level. Table 1, Table 2 and Table 3 list the methods of isolation, media used for isolation and identification, and three main species or genera of fungi isolated from libraries. The tables present the identified fungal taxa collected using different sampling approaches (air samples, surface, and dust as well as from book papers exhibiting evidence of fungal biodeterioration); sampling method (settle plates, vacuum samples, swabs, etc.); analytical tools (morphological or PCR identification); and the countries where the studies were carried out. The majority of the published reports used microscopic and morphological characteristics to identify the molds; however, some of the more recent studies complemented or substituted these traditional taxonomic tools with molecular analysis using PCR, with significant dominance of ITS region amplification.

Based on the data reviewed, we note that passive air sampling remains a common sampling method, even with the availability of many brands of air samplers. The sterile swab technique dominates surface, dust, and book sampling, whereas some researchers have used less common methods, such as media strips, to collect samples. In regard to fungal identification, culture-based microscopic/morphological analysis is still the preferred choice for most researchers. However, some groups performed hybrid analyses by combining classical microbiological methods and molecular techniques.

Table 1 lists 21 studies in which fungi were isolated from library air samples. In ten of these studies, settle plates (also known as sedimentation plates) were simply exposed to library air. This method is commonly used in hospital clean rooms to assess the level of microorganisms. The remainder of these studies used some kind of sampling pump. *Aspergillus* was the most common genus found in ten studies from samples collected in Colombia, India, Indonesia, Mexico, Turkey, and the USA using both settle plates and pumps; it was the second most common when air samples were taken from libraries in Bangladesh and India. In five studies, with sampling performed in Ethiopia, Greece, Pakistan, and Poland, *Cladosporium* was the most common genus and the second most common genus in an additional four studies from Brazil, India, and Indonesia (Table 1). The third most common genus isolated was *Penicillium*, dominating air samples from Brazil collected with settle plates, and from Italy, using an air sampler. Other genera found less commonly from air samples included *Alternaria*, *Curvularia*, *Chaetomium*, *Mucor*, and *Rhizopus*. The hybrid study by Vittal and Glory [81] from India used both settle plates and an air sampler. *Aspergillus* was the most common genus isolated using both methods. With settle plates, the next most common taxa were *Penicillium* and *Cladosporium*, but with a Rotorad sampler, *Cladosporium* and *Nigrospora* were the next most prevalent [81].

We found eleven studies that identified the main genera of molds isolated directly from dust, book surfaces, and cultural heritage materials. They are listed in Table 2. In five of the ten studies from Egypt, Indonesia, Iran, Iraq, and Slovakia, *Aspergillus* was the most common genus isolated; *Cladosporium* was the most common in three of them. The report from Morocco by El Bergdi et al. [105] found that *Penicillium* was the most common genus, while the report from Portugal, which focused on dark paper spots, found that *Chaetomium* was the most common genus. *Penicillium* was the second most common genus in nine reports that span isolations in Brazil, Egypt, Indonesia, Iran, Italy, Iraq, Portugal, and Slovakia (Table 2).

Table 3 lists 21 studies in which fungi were isolated from a combination of air, dust, and surface samples of books collected from libraries. In 13 of the 21 studies, *Aspergillus* was the most common genus isolated. *Aspergillus* was found to be the dominant genus in Brazil, Cameroon, Egypt, India, Malaysia, Nigeria, Poland, Romania, and Turkey (Table 3). (Note: *Eurotium halophilicum* is the perfect (sexual) name of a fungus now called *Aspergillus halophilicum* [132] and was the most common isolate in one of the studies from Italy. Because the Micheluz et al. [125] publication lists this species under its perfect name, we have chosen to retain this usage in the table). *Cladosporium* was the most commonly isolated in three other studies from Iran, Italy, and Poland. *Eurotium* was the most common in the study from Italy [126]. Of particular interest was the isolation of *Cryptococcus,* a human pathogen, as the second most common genus found in a Brazilian library by Leite et al. [128]. *Aspergillus* was the second most common genus in three studies; *Penicillium* was the second most common genus in ten studies, and *Cladosporium* was the second most common in three studies. Other genera that were identified included *Alternaria*, *Candida*, *Curvularia*, *Fusarium*, *Halophilicum*, *Microsporum*, *Mucor*, *Onychocola*, *Stachybotrys*, *Trichothecium*, and *Ustilago*.

Of the fifty-two studies included in our review (Table 1, Table 2 and Table 3), fourteen incorporated molecular methods, primarily PCR and, more specifically, ITS region amplification and sequencing, to identify fungi at the species level. Specifically, four studies in Table 1, seven in Table 2, and three in Table 3 used molecular and high-throughput techniques in their research. The single most common species of *Aspergillus*, identified using either traditional or molecular methods, was *A. niger* found in India, Mexico, Nigeria, Poland, Turkey, and Egypt. Similarly, several species of *Cladosporium* species were the most commonly isolated taxa from libraries sampled in Italy, Pakistan, and Greece, with *Penicillium chrysogenum* common in four studies from Egypt, Morocco, Poland, and Turkey. *Chaetomium globosum* was the most common fungal species identified in a study from Portugal by Sequeira et al. [109]. *Aspergillus* species were the second most isolated in three studies, while *Penicillium* species were the second most common in nine studies.

## 6. Discussion

Most experts recommend that surface analysis be combined with air sampling to accurately assess the types of molds that contaminate indoor environments [133]. Methods for sample collection of indoor molds and perspectives on the interpretation of these findings have been covered by Macher [28], Horner [134], Portnoy et al. [57], and Cabral [135]. Since reliance on a single sampling method does not provide an accurate estimate of fungal contamination in each environment, it has been recommended that multiple sampling techniques should be used [43]. All sampling methods based on cultivation detect only viable fungal propagules. In the future, we hope that PCR will become more widely used. PCR has the additional advantage of providing species identification in most cases.

There is a lack of standardized protocols for assessing molds from environmental samples. Thus, the published reports about mold in libraries are uneven in size, scope, methodology, and rigor. Furthermore, microbial ecologists and other experts regularly emphasize that culturable organisms comprise a small fraction of the total number of microbes in each environment. That said, there was a remarkable similarity in the major genera of viable fungi isolated from libraries across the world, with *Aspergillus*, *Cladosporium*, and *Penicillium* being the dominant genera. *Cladosporium* spores often dominate outdoors in atmospheric bioaerosols [136,137], while it is common for spore counts of *Aspergillus* and *Penicillium* species to be higher indoors than outdoors [40]. Decades of research have shown that *Acremonium*, *Aspergillus*, *Penicillium*, *Cladosporium*, *Stachybotrys*, *Ulocladium*, *Arthrinium*, *Aureobasidium*, and *Mucor* are other major genera reported from indoor environments [138]. In a broad study of damp and water-damaged building materials, Anderson et al. [139] showed that the most common genera of fungi detected were *Acremonium*, *Aspergillus*, *Chaetomium*, *Cladosporium*, *Penicillium*, and *Ulocladium*. At the species level, the most common taxa were *Aspergillus versicolor* and *Penicillium chrysogenum* [139].

These published studies on fungi in libraries have focused on “source characterization”, i.e., the identification of the culturable fungi that cause biodeterioration. Most studies did not address approaches for remedial action, and there has been almost no research on human health assessments in library settings or on the most effective strategies for controlling and preventing mold growth inside library buildings. Nevertheless, it is of concern that *A. fumigatus* [79,104] and *A. flavus* [87,92], both of which can cause aspergillosis, were each found twice as the most common species found in libraries [140].

Finally, it should be emphasized that scientific agreement is lacking for the single best protocol for measuring fungi in indoor air, nor is there a scientific consensus on the health implications to related to their presence [20,57,141]. Currently, there are no universally accepted guidelines or national standards that define safe levels of human exposure to mold spores, mycelial fragments, bacteria, and other microbial bioaerosols. As stated by scientists at the USA “National Institutes of Occupational Safety and Health (NIOSH)”: “There are no health-based standards for acceptable levels of biological agents in indoor air” [142].

As different commercially available bioaerosol samplers have been shown to considerably affect results, consistent monitoring techniques are necessary to ensure that data are accurate and comparable. The first standardized method was published by NIOSH in 1998, followed by a series of indoor air standards, including mold sampling, implemented by the International Organization for Standardization (ISO) after research and extensive testing in numerous laboratories [143]. As of right now, this organization has released 23 standards [144]. The international ISO 16000 set of indoor standards has been the pioneer in the development of indoor air data quality guidelines and standards [145]. ISO 16000-1 establishes general requirements relating to the measurement of indoor air pollutants and the important conditions to be observed before or during sampling. Aspects of identification and sampling strategy for specific pollutants or groups of pollutants are specified in subsequent parts of ISO 16000. ISO 16000-5, which deals with the VOC sampling strategy, is a link between ISO 16000-1 and this part of ISO 16000, which deals with sampling and analytical measurements [146].

## 7. Summary and Conclusions

Mold growth in libraries is a worldwide problem affecting libraries in all countries regardless of geographic location. When library materials become wet, molds can grow. Most of the published studies that study fungal growth in libraries use source sample methods such as swabs and tape lifts, either alone or combined with an air sampling approach. The samples are then viewed under a microscope or cultured on laboratory media for further identification. A few recent studies have circumvented the problems associated with non-viable spores by using PCR. In the 53 studies reviewed, the location of sampling sites, sampling methods, media used for culturing, and number of samples taken varied extensively among studies. In all approaches, *Aspergillus*, *Penicillium*, and *Cladosporium* were the most abundant genera in library environments. The survey from Brazil by Leite et al. [129] is the only one that found *Cryptococcus*, a dangerous human fungal pathogen that affects immunocompromised people. Acceptable levels of indoor air contamination with molds and other microorganisms have not been established due to the difficulty in creating reference values, but building scientists and microbiologists agree that it is prudent to remediate mold damage as soon as possible to minimize damage to inanimate materials and jeopardy to human health.

Jorge Luis Borges, the twentieth-century Argentine writer, is best known for his books and poems about dreams, labyrinths, and archives. It is often forgotten that he spent part of his youth as a librarian and that later in life, between 1955 and 1973, he was director of the “Argentine National Library”. Borges once said that “if I were asked to name the chief event in my life, I should say my father’s library”. We have chosen a phrase from Borges’ famous quote about paradise and libraries for our title because we believe that after fires, mold is the most important risk to the preservation of library materials. Unlike fires, which have received widespread attention from historians, the insidious, quiet destruction caused by filamentous fungi has been largely ignored. When libraries do suffer water damage, librarians often consult local mycologists and microbiologists for their help, most of whom are not trained to study indoor air. We hope that our review will guide scientists who are called in to give advice to librarians and increase general awareness about the role molds play in the destruction of books, archives, and other cultural heritage materials.

## Figures and Tables

**Figure 1 jof-09-01061-f001:**
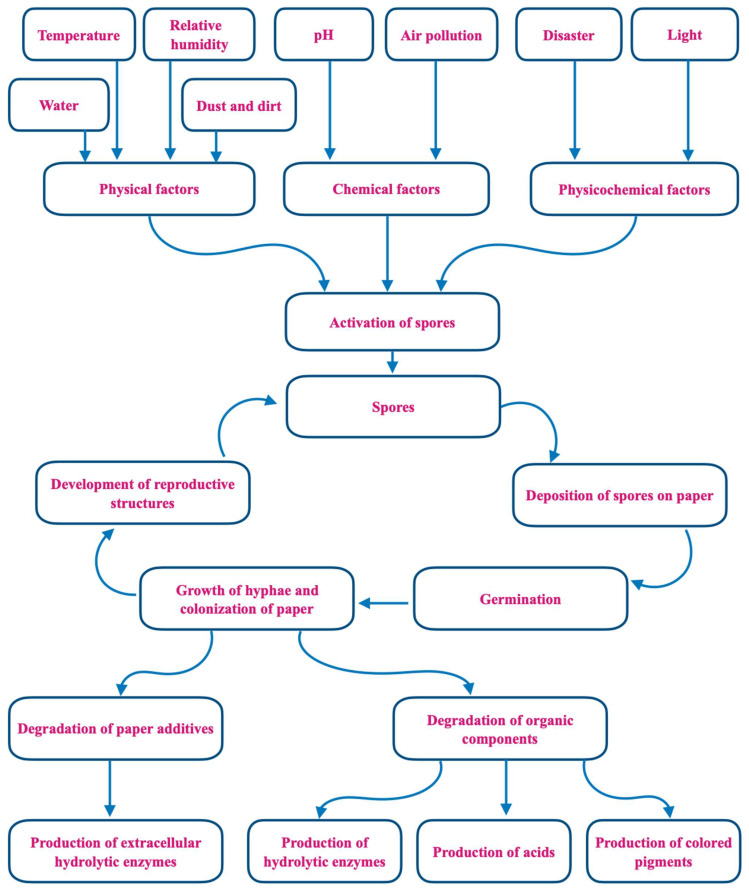
Mechanisms of fungal biodeterioration of paper.

**Figure 2 jof-09-01061-f002:**
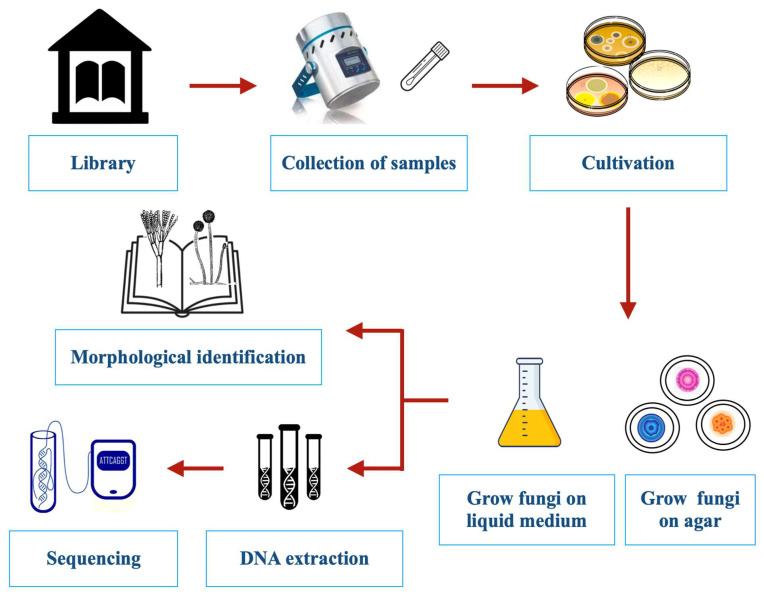
Experimental protocols for the detection, isolation, and identification of fungi colonizing libraries.

**Figure 3 jof-09-01061-f003:**
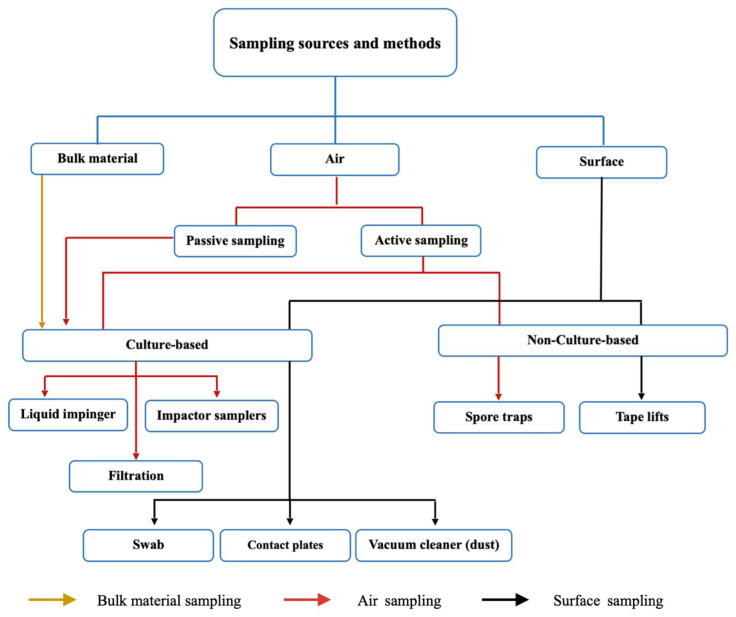
Conventional sampling strategies for isolating, enumerating, and identifying fungi from indoor environments.

**Table 1 jof-09-01061-t001:** Main genera of fungi identified from library air samples.

Geographic Location of Library/Collection Method	Culture Media ^a^/Identification Method ^b^	Main Taxa of Fungi Detected ^c^	Reference
**USA** Andersen volumetric viable particle samplers	MEA and FPA	*Aspergillus fumigatus*, *Aspergillus niger*, and *Cladosporium herbarum*	[79]
**POLAND** Settles plates	CZA, YEA, and SDA	*Cladosporium herbarum*, *Penicillium chrysogenum*, and *Aspergillus niger*	[80]
**INDIA** Rotorod sampler and settle plates	PDA and SDA	**Rotorod sampling:** *Aspergillus* spp., *Cladosporium*, and *Nigrospora* **Plate sampling:** *Aspergillus* spp., *Curvularia*, and *Penicillium*	[81]
**ITALY** Settles plates; personal volumetric air sampler spore trap	CDA and SDA	*Alternaria* sp., *Chaetomium* sp., and *Cladosporium* sp.	[82]
**BANGLADESH** Settle plates	Hemolytic activity test using blood agar	*Alternaria*, *Aspergillus*, and *Curvularia*	[83]
**INDIA** Settle plates	CZA and PDA	*Aspergillus niger*, *Aspergillus fumigatus*, and *Curvularia*	[84]
**INDIA** Buck Bioculture pump	PDA, EMA, and blood agar	*Rhizopus oryzae*, *Aspergillus nidulans*, and *Aspergillus flavus*	[85]
**MEXICO** Microbio MB2^®^Aerosol Sampler	PDA **PCR (ITS1–5.8S–ITS2)**	*Aspergillus niger*, *Aspergillus tamarii*, and *Aspergillus oryzae*	[86]
**INDIA** Settle plates	PDA	*Aspergillus flavus*, *Cladosporium cladosporioides*, and *Mucor pusillus*	[87]
**ETHIOPIA** Settle plate	SDA	*Cladosporium* sp., *Alternaria* sp., and *Penicillium* sp.	[88]
**INDIA** Rotorod air sampler	Spore counts	*Aspergillus*, *Cladosporium*, and *Alternaria*	[89]
**INDIA** Settle plates	SDA and PDA	*Aspergillus niger*, *Aspergillus flavus*, and *Aspergillus fumigatus*	[90]
**ITALY** Active Surface Air System (plate impact active sampler)	SDA **PCR and NGS sequencing**	*Penicillium rivolii*, *Penicillium viticola*, and *Cladosporium uredinicola*	[91]
**TURKEY** Portable volumetric microbiological air sampler	DG18, MEA, CDA, CY20S, CYA, and PDA	*Aspergillus flavus*, *Penicillium chrysogenum*, and *Alternaria alternata*	[92]
**INDONESIA** Settle plates	DG 18 for sampling; MEA, DG 18, or CY20S for identification	*Aspergillus*, *Cladosporium*, and *Penicillium*	[93]
**INDIA** Air sampler: Hi-Air sampler	Two media strips:PS-640 and PS-290	*Curvularia lunata*, *Curvularia geniculata*, and *Curvularia tetramera*	[94]
**GREECE** Single-stage Portable Burkard sampler	MEA **PCR (ITS1-5.8S-ITS2, b-tubulin and calmodulin)**	*Cladosporium*, *Penicillium*, and *Aspergillus*	[95]
**BRAZIL** Settle plates	Collection SDA, PDA, and SDA + chloramphenicol for isolation	*Penicillium* sp., *Cladosporium* sp., and *Alternaria* sp.	[96]
**PAKISTAN** Air sampler (Gilian5000)	PDA **PCR (ITS1 and ITS4)**	*Cladosporium asperulatum*, *Penicillium oxalicum*, and *Aspergillus niger*	[97]
**COLOMBIA** Two-stage cascade impactor	SDA	*Aspergillus*, *Curvularia*, and *Cladosporium*	[98]
**POLAND** Settle plates, six-stage Andersen cascade sampler	SDA	*Cladosporium*, *Penicillium*, and *Aspergillus* spp.	[99]

**^a^** CDA: Czapek–Dox agar; CY20S: Czapek yeast extract agar + 20% sucrose; CYA: Czapek yeast autolysate agar; CZA: Czapek agar; DG18: Dichloran 18% glycerol agar; EMA: eosin methylene blue agar; FPA: filter paper afar; MEA: malt extract agar; PDA: potato dextrose agar; SDA: Sabouraud dextrose agar. YEA: yeast extract agar. **^b^** Fungi were identified using traditional cultural, morphological, and microscopic characters unless otherwise noted. **^c^** Listed in order of frequency. Some taxa were identified only to genus.

**Table 2 jof-09-01061-t002:** Main genera of fungi isolated directly from surface samples of dust, book surfaces, and cultural heritage materials preserved in libraries.

Geographic Location of Library/Collection Method	Culture Media ^a^/Identification Method ^b^	Main Taxa of Fungi Detected ^c^	Reference
**IRAN** Sterile swabs and scalpels	SDA + Chloramphenicol	*Aspergillus* sp. and *Penicillium* sp.	[100]
**ITALY** Sterile cotton swabs and needles; 3 M adhesive tape	Agar and broth culture MEA, CYA, and CZA broth	*Cladosporium cladosporioides*, *Penicillium pinophilum*, and *Aspergillus versicolor*	[101]
**BRAZIL** Sterile swabs	SDA for collection; SDA and PDA for isolation; CMA, CZA, and PDA for identification	*Cladosporium*, *Penicillium*, and *Aspergillus*	[102]
**ITALY** Sterile swabs and nitrocellulose membranes; removable transparent adhesive tape	MEA, CY20S, and DG18 **PCR (ITS)**; total DNA	*Cladosporium*, *Penicillium* spp., and *Aspergillus* spp.	[103]
**SLOVAKIA** Adhesive tape and swabs	DRBC, SDA, and MEA **28S rDNA sequencing (NL1 and NL4)**	*Aspergillus fumigatus*, *Pen-icillium funiculosum*, and *Mucor spinosus*	[104]
**MOROCCO** Sterile swabs, scalpel	MEA, LB agar, YPG antibiotics agar, and screening on CMC **PCR (ITS1, 5.8S,** **and ITS2)**	*Penicillium chrysogenum*, *Aspergillus niger*, and *Mucor racemosus*	[105]
**INDONESIA** Adhesive tape and sterile swabs	PCA, PDA **ITS (ITS5-ITS4)**	*Aspergillus awamori*, *Penicillium citrinum*, and *Pseudocercospora chiangmaiensis*	[106]
**IRAQ** Sterile swabs	SDA	*Aspergillus* spp., *Penicillium* spp., *and Rhizopus* spp.	[107]
**PORTUGAL** adhesive tape (2.25 mm^2^); scalpel, tweezers, and cotton swabs	Microscopic examination of stains; PDA and MEA **PCR (ITS4** **and ITS1F)**	*Chaetomium globosum*, *Penicillium chrysogenum*, and *Myxotrichum deflexum*	[108]
**EGYPT** Sterile swabs	CYB, CYA **PCR (ITS1-ITS2)**	*Penicillium citrinum*, *Aspergillus ustus*, and *Penicillium chrysogenum*	[109]
**EGYPT** Sterile swabs, collection of deteriorated fragments	CYB, CYA **PCR (ITS1-ITS3)**	*Aspergillus niger*, *Penicillium chrysogenum*, and *Aspergillus quadrilineatus*	[110]

**^a^** BLA: banana leaf agar; CLA: carnation leaf agar; CMA: cornmeal agar; CMC: carboxymethylcellulose agar; CYA: Czapek yeast agar; CYB: Czapeck yeast extract broth; CZA: Czapek agar; DRBC: dichloran Rose Bengal chloramphenicol; LB: lysogeny broth; MEA: malt extract agar; PCA: plate count agar; PDA: potato dextrose agar; SDA: Sabouraud dextrose agar. **^b^** In most cases, fungi were identified using traditional cultural, morphological, and microscopic characteristics, and studies that used nucleic acid analysis/PCR are presented in bold. **^c^** Listed in order of frequency. Some taxa were identified only to genus.

**Table 3 jof-09-01061-t003:** Main genera of fungi isolated from air, dust, and surface samples of books collected from libraries.

Geographic Location of Library/Collection Method	Culture Media ^a^/Identification Method ^b^	Main Taxa of Fungi Detected ^c^	Reference
**LITHUANIA** Slit-to-agar single-stage impactor (Krotov 8180; settle plates; sterile swabs and adhesive tape)	MEA and SDA for sampling; CZA, CMA, and MEA for identification	*Penicillium expansum*, *Aspergillus flavus*, *and Mucor* spp.	[111]
**ROMANIA** Settle plates and sterile swabs	PDA and MEA for isolation; CDA, MEA, PDA, CGA, and SDA for identification	*Aspergillus* spp., *Penicillium* spp., and *Cladosporium* spp.	[112]
**POLAND** Impaction method; press counting plate with MEA against artifact surface	MEA for sampling; CZA for identification	*Cladosporium* and *Penicillium*	[113]
**POLAND** Six-stage Andersen Impactor; GSP and Button aerosol samplers; sterile swabs	MEA	*Aspergillus niger*, *Penicillium verrucosum*, and *Candida famata*	[114]
**POLAND** Six-stage Graseby–Andersen impactor	MEA	*Penicillium* spp., *Trichothecium laxicephalum*, and *Alternaria tenuis*	[115]
**PORTUGAL** M Air Tester, sterile swabs	MEA, DG18, and MA **28S and ITS**	*Penicillium* spp. and *Cladosporium* spp.	[116]
**BRAZIL** Settle plate and sterile swabs	SDA for isolation; MEA and CZA for identification	*Aspergillus* spp. and *Penicillium* spp.	[117]
**ITALY** DUO SAS 360 sampler; vacuum cleaner Settle plates; Hirst spore trap; volumetric trap; nitrocellulose filters	SDA	*Cladosporium* spp., *Fusarium* spp., and *Ustilago* spp.	[118]
**IRAN** Settle plates; sterile swabs	SDA, yeast colonies tested with germ tube and chlamydoconidia production	*Cladosporium* sp., *Penicillium* sp., and *As-pergillus* sp.	[119]
**TURKEY** Settle plates; sterile swabs	RbCA, PDA, MEA, CDA	*Aspergillus niger*, *Penicillium*, and *Cladosporium herbarum*	[120]
**POLAND** Air sampler MAS-110 Ec; six-stage Andersen sampler; swab method using saline solution	TSA, DG18 and MEA for air samples; SDA for surface samples; CYA and YES for identification **PCR (ITS1/2)**	*Aspergillus puulaauensis*, *Cladosporium cladosporioides*, and *Penicillium crustosum*	[121]
**MALAYSIA** Settle plate and sterile swabs	PDA and SDA	*Aspergillus* sp., *Onychocola* sp., and *Microsporum* sp.	[122]
**INDIA** Settle plates; cotton swabs; sections of deteriorated paper	PDA; analysis of cellulase activity	*Aspergillus* spp. and *Fusarium*	[123]
**MALAYSIA** Coriolis air sampler, settle plates, sterile swabs	MEA	*Aspergillus* sp., *Penicillium* sp., and *Stachybotrys* sp.	[124]
**ITALY** Sterile swabs, adhesive tape, Sampl’Air Lite sampler	MEA	*Eurotium halophilicum* and *Aspergillus creber.*	[125]
**TURKEY** Portable volumetric microbiological air sampler; sterile swabs	DG18, MEA, PDA MF were inoculated into CDA, CY20S, CYA, MEA, PDA, and 25% glycerol nitrate agar	*Aspergillus versicolor*, *Cladosporium sphaerospermum*, and *Penicillium dierckxii*	[126]
**EGYPT** Andersen two-stage air sampler	Rose Bengal streptomycin agar for sampling; SDA, MEA, and CDA for identification Analysis of fungal enzymatic activity	*Aspergillus niger*, *Aspergillus fumigatus*, and *Penicillium*	[127]
**BRAZIL** Andersen air sampler; settle plates; sterile swabs	SDA for isolation; CYA 25, CYA37, CY20S, MEA CLA, BLA, and PDA for identification	*Aspergillus niger*, *Cryptococcus*, and *Cladosporium cladosporioides*	[128]
**NIGERIA** Settle plates; nitrocellulose membranes	MEA; focus on cellulolytic isolates **PCR (NS5 and ITS4)**	*Aspergillus niger* and *Penicillium georgiense*	[129]
**CAMEROON** Settle plate and sterile swabs	SDA	*Aspergillus* spp., *Penicillium* spp., and *Curvularia* spp.	[130]
**ROMANIA** Destructive and nondestructive methods; Settle plates; Air sampler	DG 18	*Penicillium* spp., *Cladosporium* spp., and *Fusarium* spp.	[131]

**^a^** CGA: chloramphenicol glucose agar; CDA: Czapek–Dox agar; CMA: cornmeal agar; CY20S: Czapek yeast extract agar + 20% sucrose; CYA: Czapek yeast agar; CZA: Czapek agar; DG 18: dichloran 18% glycerol agar; MA: mycobiotic agar; MEA: malt extract agar; PDA: potato dextrose agar; RbCA: Rose Bengal chloramphenicol agar; SDA: Sabouraud dextrose agar; TSA: tryptic soy agar; YES: yeast extract with supplements. **^b^** In most cases, fungi are identified using traditional cultural, morphological, and microscopic characters. Studies that used nucleic acid analysis/PCR are presented in bold. **^c^** Listed in order of frequency. Some taxa were identified only to genus.

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
