# Peer review of "Mold in Paradise: A Review of Fungi Found in Libraries"

_jof, 2023, doi:10.3390/jof9111061_

Round 1

Reviewer 1 Report

The authors in this review have discussed the worldwide problem of mold growth in the libraries, which can occur when library materials are exposed to temperature changes, water, and improper ventilation. They have provided a extended literature where people have used various sampling methods, including swabs, tape lifts, air sampling and PCR. Studies have established that Aspergillus, Penicillium and Cladosporium are the most common genera in the library environments. Overall, this review highlights the importance of addressing mold change promptly to minimize the harm to material and human health. It is a well-documented review where authors have covered all the important aspects of the mold issues in the library. This review definitely provides a comprehensive view of the issue and the challenges which need to be addressed in the future.

Minor Comments:

1.    Did authors notice a pattern of fungal growth around the world i.e. fungal species more prevalent in certain parts of the world.

2.    It would be good if authors could add a section where they describe the sampler methods more in detail i.e., Andersen, Burkard etc.

3.    Instead of discussing the remedial issues in the discussion it would be good if they could make a subsection on this.

4.    There are a few grammatical and space errors which needs to be addressed. 

MODERATE CHANGES WITH GRAMMER

Author Response

Comments and Suggestions for Authors

The authors in this review have discussed the worldwide problem of mold growth in the libraries, which can occur when library materials are exposed to temperature changes, water, and improper ventilation. They have provided a extended literature where people have used various sampling methods, including swabs, tape lifts, air sampling and PCR. Studies have established that Aspergillus, Penicillium and Cladosporium are the most common genera in the library environments. Overall, this review highlights the importance of addressing mold change promptly to minimize the harm to material and human health. It is a well-documented review where authors have covered all the important aspects of the mold issues in the library. This review definitely provides a comprehensive view of the issue and the challenges which need to be addressed in the future.

Minor Comments

1.    Did authors notice a pattern of fungal growth around the world i.e. fungal species more prevalent in certain parts of the world.

Authors’ response: Yes, we were alert to possible geographic differences in fungal distribution. In fact, in preparing this manuscript, in an early organization of our review data, we used geography rather than sampling method as an organization rubric. However, it soon became  apparent that published papers on mold in libraries more likely reflected the economic infrastructures supporting libraries in different countries rather than the different climates of the different countries. We reviewed the results of our literature survey and were able to reveal a relative pattern based on geographic location.

In as much as we could discern a pattern, three genera of fungi --. Aspergillus, Cladosporium and Penicillium - have been found to dominate in libraries. Aspergillus was commonly isolated from libraries in countries in Asia, Africa, and North and South America. Cladosporium was the most abundant genus in Europe while Penicillium was the second most common in African and European countries.

However, there are many reasons why it is difficult to draw valid conclusions about geographic distribution of mold contamination, The number of studies differs from one continent to another. For example, 20 searches were conducted in Asia and 19 in Europe, while only 7 were carried out in Africa, 5 in South America, and 2 in North America. Within the same continent, the distribution of the number of studies at the country level also was irregular. In South America, for example, 4  of the  5 studies published studies were conducted in Brazil. In Europe, libraries in Italy and Poland were dominant with 6 studies from each of these countries. However, we were unable to find any published studies from Scandinavian or Western European countries. Similarly, Indian studies preponderated Asia, with no sampling conducted in libraries in  Central and East Asia. Additionally,  the methods used for sampling were widely divergent. Some groups relied solely on classical methods, while others used molecular techniques allowing them to identify fungi to the species level. A more reliable pattern of geographic distribution would be observed only if all studies used the same experimental design and methods with appropriately representative numbers of studies in each continent.

2.    It would be good if authors could add a section where they describe the sampler methods more in detail i.e., Andersen, Burkard etc.

Authors’ response: We have added a new section. All the sampling techniques have been developed in detail in subsection 2.2 entitled “Sampling methods”.

3.    Instead of discussing the remedial issues in the discussion it would be good if they could make a subsection on this.

Authorsresponse: We have followed this recommendation. A new subsection (subsection 3) entitled “Mold prevention in libraries” has been added to the text as suggested. In this subsection we have discussed prevention methods for controlling mold growth in libraries, including a section about the issues facing their implementation.

4.    There are a few grammatical and space errors which needs to be addressed. 

Authors’ response: Thank you for drawing our attention to these errors. We carefully have proofread the entire manuscript and have done our best to  correct all grammatical and spatial errors.

Reviewer 2 Report

Nice piece of literature. informative and well-structured/relevant for conservationists. In my opinion, you left out a paper (Pinheiro et al. 10.1016/j.ibiod.2011.02.008) that I would like to see added to the review of 39 papers (should be added and the entire paper checked for any necessary changes resulting in adding this paper) and also to tables 1 and 3. No further recommendations. Congratulations on your work. 

Author Response

Nice piece of literature. informative and well-structured/relevant for conservationists. In my opinion, you left out a paper (Pinheiro et al. 10.1016/j.ibiod.2011.02.008) that I would like to see added to the review of 39 papers (should be added and the entire paper checked for any necessary changes resulting in adding this paper) and also to tables 1 and 3. No further recommendations. Congratulations on your work. 

Authors’ response: Thank you very much for your positive feedback. We have added the study by Pinheiro et al. (2011) to Table 3 as you requested.

Reviewer 3 Report

Islam El Jaddaoui et al. present a review manuscript entitled “Mold in paradise: a review of fungi found in libraries“, for possible publication in JOF.

The topic of the review is of high interest to a large audience since moulds are major contaminants of indoor environments and biodeteriogenic agents of many materials. Moreover, there are only a few reviews published on this subject, particularly concerning libraries.

The manuscript is well written, easy to read, but it is too superficial and needs improvements in text and illustrations.

The authors do not refer to the ISO 16000 standards which describe approaches for sampling, detection and counting of moulds. They must talk about it even if these standards are not generally used as they are in published scientific studies.

The title may be misleading as the authors focused mainly on studies that isolated and identified moulds through cultural approaches. Other experimental approaches exist to detect and characterize fungal contamination such as the analysis of volatile organic compounds, Fourier transform infrared spectroscopy, or scanning electron microscopy-EDS. some of these techniques are barely touched on in the manuscript, others are totally absent. Developing these approaches in the manuscript would enrich it.

it would also be wise to further develop the specificities of the libraries environment compared to other indoor environments (rooms welcoming the public, reserves, materials, management of air and exchanges with the outside, temperature, light, humidity, etc.). The authors must develop how these specificities are favourable to fungal development.

Another weakness of this review is its lack of illustrations. Images illustrating processes of fungal deterioration of materials in libraries, experimental protocols for fungal detection/isolation/identification, moulds isolated on culture medium would increase the impact of the manuscript.

Specific comments:

- Introduction third paragraph. Discoloration of materials is another red flag of fungal development.

- Figure 1. The colour code used must be defined in the legend. There are gaps, some words are cut off: sam-, im-? Change “Rodac“ which is a commercial word with “contact plates“.

- Develop this statement: “Building characteristics, season of the year and other environmental parameters can affect the data collected”.

- Part 2. Sampling techniques are not described in the text. The same goes for identification techniques (molecular, culture and macroscopic and microscopic observations, biochemical characters). They must be detailed and compared.

- Part 4 and before: Beyond the fungal genus, the identification of the species can be important, in particular for the health risk. Indeed, some species such as Aspergillus fumigatus are endowed with a particular infectious potential in humans, others such as Stachybotrys chartarum produce very dangerous mycotoxins.

- Part 5 and before: Details on the sequences targeted by the PCR (ITS, 18S RNA genes, others) and the associated sequencing techniques must be given.

Here is a non-exhaustive list of standards and articles likely to contribute to enriching the review:

- ISO 16000-16, ISO 16000-17, ISO 16000-18, ISO 16000-19, ISO 16000-20, ISO 16000-21 (sampling, detection, enumeration of moulds, spore count).

- Nevalainen A, Täubel M, Hyvärinen A. Indoor fungi: companions and contaminants. Indoor Air. 2015 Apr;25(2):125-56. doi: 10.1111/ina.12182.

- Bronislaw Zyska, Fungi isolated from library materials: A review of the literature,

International Biodeterioration & Biodegradation, Volume 40, Issue 1, 1997, 43-51, https://doi.org/10.1016/S0964-8305(97)00061-9.

- Clara Urzı̀, Filomena De Leo, Sampling with adhesive tape strips: an easy and rapid method to monitor microbial colonization on monument surfaces, Journal of Microbiological Methods, Volume 44, Issue 1, 2001, 1-11, https://doi.org/10.1016/S0167-7012(00)00227-X.

- Malalanirina S. Rakotonirainy, Olivier Bénaud, Léon-Bavi Vilmont, Contribution to the characterization of foxing stains on printed books using infrared spectroscopy and scanning electron microscopy energy dispersive spectrometry, International Biodeterioration & Biodegradation, Volume 101, 2015, 1-7, https://doi.org/10.1016/j.ibiod.2015.02.031.

- A. Lecellier, V. Gaydou, J. Mounier, A. Hermet, L. Castrec, G. Barbier, W. Ablain, M. Manfait, D. Toubas, G.D. Sockalingum, Implementation of an FTIR spectral library of 486 filamentous fungi strains for rapid identification of molds, Food Microbiology, Volume 45, Part A, 2015, 126-134, https://doi.org/10.1016/j.fm.2014.01.002.

- Jose Ruiz-Jimenez, Sanni Raskala, Ville Tanskanen, Elisa Aattela, Mirja Salkinoja-Salonen, Kari Hartonen, Marja-Liisa Riekkola, Evaluation of VOCs from fungal strains, building insulation materials and indoor air by solid phase microextraction arrow, thermal desorption–gas chromatography-mass spectrometry and machine learning approaches,

Environmental Research, Volume 224, 2023, 115494, https://doi.org/10.1016/j.envres.2023.115494.

Latgé JP, Chamilos G. Aspergillus fumigatus and Aspergillosis in 2019. Clin Microbiol Rev. 2019 Nov 13;33(1):e00140-18. doi: 10.1128/CMR.00140-18.

- Dyląg M, Spychała K, Zielinski J, Łagowski D, Gnat S. Update on Stachybotrys chartarum-Black Mold Perceived as Toxigenic and Potentially Pathogenic to Humans. Biology (Basel). 2022 Feb 23;11(3):352. doi: 10.3390/biology11030352.

Minor editing of English language required.

Author Response

Islam El Jaddaoui et al. present a review manuscript entitled Mold in paradise: a review of fungi found in libraries, for possible publication in JOF.

The topic of the review is of high interest to a large audience since moulds are major contaminants of indoor environments and biodeteriogenic agents of many materials. Moreover, there are only a few reviews published on this subject, particularly concerning libraries. The manuscript is well written, easy to read, but it is too superficial and needs improvements in text and illustrations.

The authors do not refer to the ISO 16000 standards which describe approaches for sampling, detection and counting of moulds. They must talk about it even if these standards are not generally used as they are in published scientific studies.

Authors’ response: As requested, we have added a paragraph about the ISO 16000 standards at the end of the discussion section.

Another weakness of this review is its lack of illustrations. Images illustrating processes of fungal deterioration of materials in libraries, experimental protocols for fungal detection/isolation/identification, moulds isolated on culture medium would increase the impact of the manuscript.

Authors’ response: Thank you for this excellent suggestion . Two additional illustrations have been added. Figure 1 illustrates the main mechanisms of fungal biodeterioration of paper while Figure 2 provides an overview of the experimental protocols for detection, isolation, and identification of fungal colonizing libraries.

The title may be misleading as the authors focused mainly on studies that isolated and identified moulds through cultural approaches. Other experimental approaches exist to detect and characterize fungal contamination such as the analysis of volatile organic compounds, Fourier transform infrared spectroscopy, or scanning electron microscopy-EDS. some of these techniques are barely touched on in the manuscript, others are totally absent. Developing these approaches in the manuscript would enrich it.

Authors’ response:  In response to this suggestion we have added considerable explanatory text.  The proposed experimental approaches have been developed and detailed in section 2.2.  Identification techniques”.

It would also be wise to further develop the specificities of the libraries environment compared to other indoor environments (rooms welcoming the public, reserves, materials, management of air and exchanges with the outside, temperature, light, humidity, etc.). The authors must develop how these specificities are favourable to fungal development.

Author’s response:  Our manuscript is already quite long. We are planning to write a future article aimed at the library science community in which we will cover some of these parameters.  Moreover, several of the resources cited in our current text provide this information for interested readers.   In summary, we hope that the editor and reviewers do not require that we discuss the specifics of library architecture, management, and so forth in this paper, which is intended as a review of published studies about mold taxa found in libraries. 

Specific comments

Introduction third paragraph. Discoloration of materials is another red flag of fungal development.

Authors’ response: Discoloration in the form of ‘foxing’  is now described in the third paragraph of the introduction as suggested.

Figure 1. The colour code used must be defined in the legend. There are gaps, some words are cut off: sam-, im-? Change Rodac“ which is a commercial word with contact plates.

Authorsresponse: All inappropriate abbreviations mentioned in the figure have been corrected and the color coding used is defined in the legend.

Develop this statement: Building characteristics, season of the year and other environmental parameters can affect the data collected”.

Authors’ response: the sentence was expanded as follows: “The season of the year, building characteristics such as construction materials, air conditioning system, ventilation and light source as well as other environmental parameters, mainly indoor temperature and relative humidity, can affect the data collected.”

Part 2. Sampling techniques are not described in the text. The same goes for identification techniques (molecular, culture and macroscopic and microscopic observations, biochemical characters). They must be detailed and compared.

Authorsresponse:  In response to this suggestion, we have added considerable text to the manuscript.  Air and surface sampling techniques are now detailed in section “2.1 Sampling Methods”. Likewise, details about  identification methods are detailed in a separate section 2.2 Identification Techniques”.

Part 4 and before: Beyond the fungal genus, the identification of the species can be important, in particular for the health risk. Indeed, some species such as Aspergillus fumigatus are endowed with a particular infectious potential in humans, others such as Stachybotrys chartarum produce very dangerous mycotoxins.

Author’s response:  The revised text now calls attention to the pathogenic fungi that were identified in several studies. These include Aspergillus fumigatus and A. flavus, as well as Cryptococcus.

Part 5 and before: Details on the sequences targeted by the PCR (ITS, 18S RNA genes, others) and the associated sequencing techniques must be given.

Authorsresponse: an entire paragraph has been added in section 2.2. Identification Techniques” detailing PCR and possible target sequences, with mention of associated sequencing approaches.

Here is a non-exhaustive list of standards and articles likely to contribute to enriching the review:

- ISO 16000-16, ISO 16000-17, ISO 16000-18, ISO 16000-19, ISO 16000-20, ISO 16000-21 (sampling, detection, enumeration of moulds, spore count).

- Nevalainen A, Täubel M, Hyvärinen A. Indoor fungi: companions and contaminants. Indoor Air. 2015 Apr;25(2):125-56. doi: 10.1111/ina.12182.

- Bronislaw Zyska, Fungi isolated from library materials: A review of the literature,

International Biodeterioration & Biodegradation, Volume 40, Issue 1, 1997, 43-51, https://doi.org/10.1016/S0964-8305(97)00061-9.

- Clara Urzı̀, Filomena De Leo, Sampling with adhesive tape strips: an easy and rapid method to monitor microbial colonization on monument surfaces, Journal of Microbiological Methods, Volume 44, Issue 1, 2001, 1-11, https://doi.org/10.1016/S0167-7012(00)00227-X.

- Malalanirina S. Rakotonirainy, Olivier Bénaud, Léon-Bavi Vilmont, Contribution to the characterization of foxing stains on printed books using infrared spectroscopy and scanning electron microscopy energy dispersive spectrometry, International Biodeterioration & Biodegradation, Volume 101, 2015, 1-7, https://doi.org/10.1016/j.ibiod.2015.02.031.

- A. Lecellier, V. Gaydou, J. Mounier, A. Hermet, L. Castrec, G. Barbier, W. Ablain, M. Manfait, D. Toubas, G.D. Sockalingum, Implementation of an FTIR spectral library of 486 filamentous fungi strains for rapid identification of molds, Food Microbiology, Volume 45, Part A, 2015, 126-134, https://doi.org/10.1016/j.fm.2014.01.002.

- Jose Ruiz-Jimenez, Sanni Raskala, Ville Tanskanen, Elisa Aattela, Mirja Salkinoja-Salonen, Kari Hartonen, Marja-Liisa Riekkola, Evaluation of VOCs from fungal strains, building insulation materials and indoor air by solid phase microextraction arrow, thermal desorption–gas chromatography-mass spectrometry and machine learning approaches,

Environmental Research, Volume 224, 2023, 115494, https://doi.org/10.1016/j.envres.2023.115494.

- Latgé JP, Chamilos G. Aspergillus fumigatus and Aspergillosis in 2019. Clin Microbiol Rev. 2019 Nov 13;33(1):e00140-18. doi: 10.1128/CMR.00140-18.

- Dyląg M, Spychała K, Zielinski J, Łagowski D, Gnat S. Update on Stachybotrys chartarum-Black Mold Perceived as Toxigenic and Potentially Pathogenic to Humans. Biology (Basel). 2022 Feb 23;11(3):352. doi: 10.3390/biology11030352.

Authorsresponse: Thank you very much for sharing this list of valuable references with us, it has helped us considerably to enrich and improve our text.

Round 2

Reviewer 3 Report

The authors improved their manuscript by correcting what needed to be corrected and enriching it according to recommendations. It is now of high quality and may be accepted for publication in JOF.

Minor editing of English required

Author Response

The authors improved their manuscript by correcting what needed to be corrected and enriching it according to recommendations. It is now of high quality and may be accepted for publication in JOF.

Minor editing of English required

Authors’ response: Thank you for this request, the English quality of the manuscript has been improved